# Digital Microfluidics in Newborn Screening for Mucopolysaccharidoses: A Progress Report

**DOI:** 10.3390/ijns6040078

**Published:** 2020-10-08

**Authors:** Jon Washburn, David S. Millington

**Affiliations:** 1Baebies, Inc., Durham, NC 27709, USA; jwashburn@baebies.com; 2Duke University Hospital Biochemical Genetics Lab, Durham, NC 27709, USA

**Keywords:** newborn screening, digital microfluidics, Hurler syndrome, alpha-L-iduronidase assay, dried blood spots

## Abstract

Newborn screening (NBS) for mucopolysaccharidosis type I (MPS I, Hurler syndrome) is currently conducted in about two-fifths of the NBS programs in the United States and in a few other countries. Screening is performed by measurement of residual activity of the enzyme *alpha*-l-iduronidase in dried blood spots using either tandem mass spectrometry or digital microfluidic fluorometry (DMF). In this article, we focus on the development and practical experience of using DMF to screen for MPS I in the USA. By means of their responses to a questionnaire, we determined for each responding program that is screening for MPS I using DMF the screen positive rate, follow-up methods, and classification of confirmed cases as either severe or attenuated. Overall, the results show that at the time of reporting, over 1.3 million newborns in the US were screened for MPS I using DMF, 2094 (0.173%) of whom were screen positive. Of these, severe MPS I was confirmed in five cases, attenuated MPS I was confirmed in two cases, and undetermined phenotype was reported in one case. We conclude that DMF is an effective and economical method to screen for MPS I and recommend second-tier testing owing to high screen positive rates. Preliminary results of NBS for MPS II and MPS III using DMF are discussed.

## 1. Introduction

Newborn screening (NBS) for lysosomal storage disorders (LSDs) has been a topic of considerable interest for over twenty years, especially in the United States (US). The first published method to show proof of principle for an LSD test applicable to dried blood spots (DBS) was an enzymatic assay for α-l-iduronidase (IDUA) developed by Chamoles et al. and applied to Hurler syndrome (MPS I) [1]. The method used reagents that released a fluorophore (4-methylumbelliferone; 4-MU) when exposed to the residual enzymes in an aqueous extract from a DBS. This method was basically an adaptation of existing methods used by biochemical genetics laboratories to measure enzyme activity in extracts from whole blood. Scaling down this method to yield a robust assay requiring just a few µL of blood from a 3 mm diameter punch from a DBS was a considerable achievement that was subsequently replicated for other LSDs, including Pompe disease [2]. Methods for LSD enzymatic assays based on tandem mass spectrometry (MS/MS) with synthetic reagents were also being developed by Gelb et al. at about this time. The first published MS/MS enzymatic method targeted Krabbe disease [3], which was soon expanded to include several other LSDs [4], including MPS I [5]. The emergence of these methods and promising new therapeutic strategies such as enzyme replacement therapy (ERT) and hematopoietic stem cell transplantation (HSCT) provided the impetus to develop high-throughput methodologies to screen newborns for LSDs using dried blood spots [6,7].

Digital microfluidics (DMF), based on the principle of electrowetting, is a method that manipulates aqueous droplets on printed circuit boards using electrical impulses [8]. Programmable droplet dispensing and mixing from reagent and sample reservoirs on a disposable cartridge facilitated a variety of assays applicable to clinical diagnostics [9]. The first exploratory application of DMF to NBS was a collaborative project between Advanced Liquid Logic, Inc., the North Carolina Public Health Laboratory, and scientists at Duke University Medical Center, published over ten years ago [10]. Shortly thereafter, a critical step forward was taken when a multiplex assay for five LSD enzymes was developed using a prototype cartridge that could process 12 NBS samples [11]. In principle, the DMF method miniaturizes and automates the aforementioned 96-well plate format fluorometric assay using reagents specific for each targeted enzyme. The first NBS pilot study using this DMF prototype was performed in the Illinois Newborn Screening Laboratory on over 8000 specimens [12]. Shortly thereafter, an improved cartridge that could process 48 specimens was developed and validated for a multiplex assay of five LSDs [13].

Newborn screening programs in the US generally follow a protocol for the addition of new conditions for NBS referred to as the recommended uniform screening panel (RUSP) that was established by a consensus committee of experts from the American College of Medical Genetics in 2006 [14]. At that time, there were 29 core conditions listed as primary targets for NBS. The ACMG Committee was later encompassed by the U.S. Department of Health and Human Services Secretary’s Advisory Committee on Heritable Disorders in Newborns and Children (ACHDNC) [15], which subsequently added to the RUSP severe combined immune deficiency (SCID) in 2009, critical congenital cyanotic heart disease (CCHD) in 2010, and more recently Pompe disease (Glycogen Storage Disease-II) in 2013, Hurler Syndrome (MPS I) in 2016, X-linked adrenoleukodystrophy (X-ALD) in 2016, and spinal muscular atrophy (SMA) in 2018, making a total of 35 primary (core) conditions as of the time of writing. Despite the late addition of the two LSDs (Pompe disease and MPS I) and the fact that the first application to add Pompe disease to the RUSP in 2008 was unsuccessful, special interest groups lobbied to add LSDs to state NBS menus and were successful in several states, including New York, Illinois, Missouri, and others. Consequently, several state programs in the US began screening for LSDs prior to their appearance on the RUSP. Mandates were issued by state legislators without due consideration for the added costs and infrastructure needed to implement NBS for LSDs. It was in this setting that both the MS/MS and DMF based multiplex LSD enzyme assays began to emerge in NBS. Currently, 22 programs in the US are performing full-population NBS for MPS I plus at least one additional LSD; about one-third of them are using DMF and the rest employ MS/MS.

This manuscript focuses on only those programs that adopted DMF as their screening method for LSDs and summarizes their collective experience with NBS for MPS I by means of a questionnaire distributed in August 2020. An overview of the principles and features of DMF that make it attractive for NBS and some new data regarding its potential to add more MPS enzyme tests are also presented.

## 2. MPS I Newborn Screening Using Digital Microfluidics

### 2.1. Platform Overview

The DMF platform originally developed by Advanced Liquid Logic consists of a disposable cartridge that is loaded with the reagents and samples for analysis and is inserted into a benchtop instrument containing electronics that automate the assay protocols. All operations within the cartridge are controlled by software loaded onto an attached computer, which is capable of controlling up to four instruments simultaneously. The cartridge consists of two plates: the lower plate is a printed circuit board (PCB) with discrete electrodes coated with a dielectric and hydrophobic material, and the top plate is a clear plastic sheet with a conductive hydrophobic coating on the inner surface. These two plates form a sandwich that is filled with an oil to prevent evaporation of the aqueous droplets. Samples and reagents are loaded through the top plate into reservoirs within the cartridge. Droplets are drawn from the reservoirs onto the PCB, attracted by the principle of electrowetting, and manipulated along paths of electrodes using electrical impulses controlled by a computer program [9]. (A brief video illustrating the movement of droplets in a DMF LSD newborn screening assay is available online at: https://bit.ly/3iinddB).

The volume of each sample or reagent droplet is determined simply by the area of the electrode adjacent to each reservoir and in this particular system is less than 200 nL, dispensed with an imprecision of <2% error.

For an enzymatic assay, a sample droplet is mixed with a reagent droplet, the mixed droplet is incubated for 1 h, the reaction is stopped by adding a droplet of quenching reagent, and the resulting droplet is moved to a fixed location where a detector reads the fluorescence from the reaction product. Finally, the droplet is moved to a waste reservoir. On the current production cartridge, the maximum number of samples is 44 (four fixed reservoirs are required for calibration) and the maximum number of enzyme reagent wells is five [11,13]. Each enzymatic reaction is independently carried out within a discrete droplet; thus, there would be up to 220 individual reactions taking place on the cartridge. It is noteworthy that sample droplets may traverse the same electrodes with minimal cross-contamination or carry-over because the moving droplets are not in direct physical contact with the electrodes [9]. The cartridge cannot be reused; it is designed to be disposable.

### 2.2. Development and Commercialization of the DMF Platform

After the pilot study in Illinois [12], which was performed as mentioned earlier with a prototype cartridge limited to only 12 sample inputs, modifications and improvements were made that led to a much more practical model for high-throughput NBS [11,13]. This product, developed by Advanced Liquid Logic in 2012, was offered to the Missouri Public Health Laboratory (MSPHL) as a practical solution to their mandate to screen for multiple LSDs and included new reagents prepared under good laboratory practice (GLP) conditions. Consisting of two workstations, each with four DMF instruments controlled by a single computer, it had the capacity to analyze 640 DBS samples per 8 h shift. It was installed in the MSPHL in less than one day, whence the program launched a full-population pilot study in January 2013 for four LSDs—Fabry, Gaucher, Pompe, and MPS I. This model subsequently was named SEEKER^®^ by the company. At the time of its deployment in Missouri, it was not an FDA approved medical device.

Advanced Liquid Logic was acquired by Illumina, Inc. in July 2013, leading to a hiatus of approximately two years in further clinical DMF assay development until a new company, Baebies Inc., was founded in late 2014 and was able to resume research and development of DMF for NBS in mid-2015. The MSPHL NBS program was fully supported during this hiatus as a condition of the acquisition. In February 2017, SEEKER became the first United States Food and Drug Administration (FDA) authorized platform to screen newborns for LSDs [16]. It was also the first DMF device to be cleared by the FDA for clinical applications. It is noteworthy that the MSPHL program has been running for over 7.5 years with minimal modifications to the originally installed platform and is the longest-running prospective screening lab for MPS I in the US. Their publications are testimony to the effectiveness and robustness of DMF for LSD NBS [17,18].

### 2.3. Procedure for Specimen Analysis on the DMF Platform

The DMF NBS workflow has been previously described [13]. Briefly, in practice, newborn screening platforms are designed to accept 3 mm diameter samples from DBS punched in 96-well plates; thus, each section of a NBS laboratory receives their samples from a central punching station in this format. When a loaded 96-well plate containing DBS punches arrives in the LSD screening section, the first step is to add extraction solvent (100 µL) to 88 wells using a multi-channel pipet (the remaining eight wells are reserved for on-cartridge calibration solutions). The plate is then placed on an orbital shaker for 30 min at 600 rpm. During the extraction, DMF cartridges are manually loaded with filler fluid and prepared reagents for each assay. Four calibrators are then loaded onto each cartridge, followed by 40 specimens and 4 controls from the 96-well plate (using a multichannel pipettor). Each 96-well plate is loaded across two DMF cartridges. The cartridges are then ready to be loaded into the SEEKER instruments to begin the analytical program. The entire process of reagent, sample and filler loading takes approximately 5 min per cartridge and the assay is completed within 3 h.

Following the completion of the DMF assay, the provided software indicates the status of the quality control specimens relative to the control ranges. It also flags the newborn screening specimens that have activity values below the cut-off specified by the laboratory. Typically, samples with abnormal values are re-punched and re-analyzed in duplicate prior to reporting as an initial screen-positive (if the mean value is below the cut-off).

### 2.4. Practical Considerations of Digital Microfluidics as a Platform for NBS

#### 2.4.1. Advantages

The features of DMF that render it a valuable addition to NBS platforms have been reviewed elsewhere [19]. Its key attributes, compared with other NBS platforms, include minimal capital outlay for the equipment and for the infrastructure to support it, virtually maintenance free operation with no service contracts, and very low power consumption. A detailed cost analysis of DMF NBS is beyond the scope of this review, however, its cost effectiveness and the simplicity of the workflow relative to MS/MS has been previously discussed [20]. In addition, the platform requires very low sample and reagent volumes; literally hundreds of assays could be performed on the extract of a single 3 mm punch from a DBS, with very little waste to be disposed of. Variable hemoglobin concentration has no effect on fluorescence values determined by DMF, which is ascribed to the very short path length in a droplet (<0.3 mm) compared with that of a microtiter plate well [19].

A simple workflow comprising of a few straightforward steps is accessible to entry-level technicians. There is little room for human error because most of the assay steps are fully automated. Due to “spatial multiplexing” on the DMF cartridge, each enzymatic assay is performed at its optimum pH and buffer conditions—No compromises are required to meet ideal performance criteria for each enzyme assay. As results are generated within 3 h of sample preparation, it is possible to review initial results, repeat any screen-positive samples, and report results within a single 8 h shift.

#### 2.4.2. Limitations

Limitations of the current SEEKER DMF platform include the requirement for two DMF cartridges per 96-well plate, each performing up to four different enzymatic assays on 40 NBS specimens and four controls. Each workstation can accommodate 160 specimens per run. In high-throughput programs, several workstations may be required to accommodate the maximum daily sample volume; alternatively, the sample load can be distributed over one or more additional periods within the working day using fewer platforms, given the relatively short run time of 3 h per cartridge. The addition of more assays to the current platform is practicable with modifications, as discussed in Section 4, however, each modification does require FDA submission and clearance prior to deployment. The current platform is not adaptable to any modifications by the end-user, except to change the cut-offs for each analyte.

## 3. Prospective Screening Results from States Using DMF

### 3.1. Overview of Survey Design and Distribution

In the United States, seven state NBS programs (Oregon, Kansas, Maryland, Michigan, Missouri, Washington, and Virginia) are currently prospectively screening for MPS I using DMF technology. Kansas is currently conducting full population pilot testing and expects to transition to live screening and reporting later in 2020; all the other states are conducting live, full population screening.

Each laboratory was sent a brief questionnaire requesting answers to six questions about the general MPS I screening results from the start of prospective screening. Laboratories were asked for the start date, number of births during the screening period, number of confirmed MPS I cases (with information about disease severity, when possible), implementation of second tier testing (2TT)—including the type of 2TT performed, and the number of screen positive results.

### 3.2. Results and Discussion

#### 3.2.1. Summary of Responses from Laboratories Using DMF for MPS I NBS 

The results from the questionnaire sent to the participating programs are summarized in Table 1.

#### 3.2.2. Review of Results and Comparison with Published Data

Overall, these programs using DMF to screen for LSDs have, to date, cumulatively screened more than 1.3 million newborns and have thus far detected only eight confirmed cases with either severe or attenuated MPS I (approximately 1 in 163,000 births, or 0.62 cases per 100,000) or an unknown phenotype. These data are in line with the expectation of detecting 0.54–1.85 cases per 100,000 [22] and confirm the fact that MPS I is indeed a rare disorder in the United States. None of the programs, including Missouri, which has been prospectively screening for over seven years, have reported a false negative result. The higher initial presumptive positive rates seen in some programs also highlight the need for second-tier testing (2TT) to reduce the number of cases reported for clinical evaluation and follow-up testing. Most of these programs do not have the capability to perform 2TT using either molecular or biochemical methods in house, and therefore outsource such testing to external laboratories. Biochemical genetics testing by analysis of glycosaminoglycans (GAGs) in DBS [23] is primarily utilized to detect severe MPS I (Hurler syndrome), which is the main target of NBS, and may be sufficiently sensitive to detect at least some cases of attenuated MPS I [24]. Molecular testing, performed either as a 2TT or after case referral for follow-up testing, appears to be the preferred option in the United States for short-term follow-up of presumptive positives, regardless of the platform used for LSD enzyme testing [25].

The initial screen positive rates for MPS I vary considerably between the programs using DMF, by almost an order of magnitude (Table 1). The rates for Oregon, Kansas, Missouri, and Washington are between 0.05–0.09% and are similar to those reported by programs using MS/MS to screen for MPS I (see the following paragraph), while those for Michigan, Maryland, and Virginia are considerably higher.

The largest reported NBS study using MS/MS to screen for four LSDs thus far was from the state of Illinois, which recorded 151 screen positives for MPS I from 219,713 newborns screened, for a screen-positive rate of 0.069% [26]. As shown in the Illinois publication as well as in a recent article summarizing the results of a pilot study for MPS I NBS in North Carolina [27], molecular testing detects a high proportion of pseudo-deficiency alleles, plus MPS I carriers and variants of unknown significance, which contribute to a high false-positive rate. In the North Carolina pilot study, which was performed by MS/MS, there were 54 screen-positive cases from a total of 62,734 NBS tests for MPS I—A screen-positive rate of 0.086%. Taiwan have been pioneers of and have had a wealth of experience in NBS for MPS I, which they have been screening for by MS/MS since 2015 [28,29]. They reported only 16 screen-positives for MPS I out of 294,196 NBS tests for a screen-positive rate of just 0.0054%. However, it should be noted that they were screening only for MPS I and, more recently, both MPS I and MPS II, under more optimal conditions than experienced by the programs in the US screening for four LSDs and in a much less ethnically diverse population. A recent pilot study in Taiwan for four LSDs using MS/MS as performed in the US NBS programs reported 34 screen positives for MPS I out of 64,148 screened newborns, for a screen positive rate of 0.053% [30], similar to that of Illinois. A report from Italy using a 4-plex MS/MS method for LSD screening reported 52 screen positives for MPS I out of 112, 446 newborns screened, for a screen positive rate of 0.046% [31]. This program emphasized the importance of second-tier biochemical testing to reduce the referral rate. In summary, these literature values indicate that a screen-positive rate of about 0.05–0.08% is typical for programs using MS/MS to screen for multiple LSDs. 

Multiple factors can contribute to the screen positive rate for MPS I, including the relative prevalence of pseudo-deficiency variants in the screened population. Multiple pseudo-deficiency alleles have been identified that cause decreased activity using artificial substrates without any evidence of altered GAG metabolism [32]. The p.A79T, p.V322E, and p.D223N pseudo-deficiency alleles, in particular, are common in newborns of African descent as well as the African-American population [33]. It is reasonable to expect, given the high prevalence of pseudo-deficiency alleles in the African-American population, that states with a higher relative proportion of African-American births may have a higher screen positive rate, but further research is needed to quantify the relationship. Other contributing factors that can vary from program to program are the cut-offs, which, if more conservative, will lead to higher screen positives and specimen degradation that can occur during transport from the birthing centers, especially during the summer months. 

#### 3.2.3. Proposed Techniques to Minimize MPS I Screen Positive Rates

As mentioned in the foregoing section, the screen positive rate for MPS I is relatively high. The screening algorithm for most programs will usually include a repeat newborn screen for borderline positive results, as well as second-tier molecular testing to reduce the referral rates. The DBS GAG assay is used by some programs as a biochemical second-tier test to quickly identify severe cases of MPS I for immediate referral and follow-up. There is no clinical urgency for the referral of attenuated MPS I cases. As mentioned in Section 3.2.2, the NBS programs contributing to this report do not have the capacity to perform GAG assays in-house. They refer DBS samples to external laboratories that perform the test for a fee, such as the Mayo Clinical Laboratories (https://www.mayocliniclabs.com/test-catalog/Clinical+and+Interpretive/65095). In the future, NBS programs that perform NBS for amino acid and acylcarnitine disorders using MS/MS are likely to improve their performance by adding second-tier tests, as reviewed by Ombrone et al. [34]. As most of these second-tier tests utilize LC-MS/MS, it should be practicable to add second tier GAG assays on the same equipment since only a few tests would be required each day. Perhaps surprisingly, none of the reporting programs in our study utilize the post-analytical CLIR tool (Collaborative Laboratory Integrated Reports, www.clir.mayo.edu) pro-actively for the purpose of reducing the referral rate [35], despite screening simultaneously for multiple LSDs using the DMF platform. Other NBS programs have successfully applied CLIR to reduce referral rates for LSDs. In the North Carolina pilot, for example, judicious application of the CLIR tool was employed to reduce the referral rate from 54 to 19, a reduction of 65%. Of these, one was confirmed as severe MPS I, 13 newborns had pseudo-deficiency alleles, three newborns had variants of unknown significance, and two had heterozygous pathogenic variants [27]. Other reports in a recent special issue: ‘CLIR applications for Newborn Screening’ in this journal confirm its value in significantly reducing referral rates [36,37]. Retrospective application of CLIR, at least, should be considered a useful exercise for programs screening for MPS I and other LSDs to determine whether initial screen positives could be reduced prior to second-tier testing.

## 4. Expansion of DMF for NBS of Other MPS Disorders

MPS II (Hunter syndrome) meets all criteria for inclusion in the RUSP and it is anticipated to be nominated for addition to the RUSP. In addition to the aforementioned NBS program in Taiwan [28,29], Illinois has been prospectively screening all newborns for MPS II for over two years using MS/MS [38], while Missouri has used a fluorometric protocol for MPS II adapted from a previously published method [39] in a microtiter plate for approximately 18 months. A manuscript describing the initial findings for MPS II NBS in Missouri is planned for publication in this special issue. The transfer of this assay from 96-well plates to the DMF SEEKER platform is under development.

Practically, all enzymatic reactions that utilize a fluorescent readout are good candidates to miniaturize on the DMF cartridge. To date, fluorometric assays for three other MPSs (MPS IIIB, IVB, and VII) have been demonstrated on the DMF cartridge [40] with excellent separation of LSD quality control high (QCH) and quality control low (QCL) DBS materials. This research was intended to show proof of principle that such assays are available, should they be required in the future. This article also demonstrated a straightforward expansion from 5 to as many as 10 enzymatic assays on the same platform, by using a second fluorophore, such as resorufin with an additional detector specific for that chromophore, enabling the multiplexing of two fluorometric assays within the same droplet. Further optimization of these new assays on the DMF format will be required before they can be released. Recent developments of DMF that may be of future interest include a new device for point-of-care diagnostic testing in the newborn nursery [41].

## Figures and Tables

**Table 1 IJNS-06-00078-t001:** Summary of MPS I NBS results from laboratories utilizing DMF.

State:	Oregon	Kansas	Maryland	Michigan	Missouri	Washington	Virginia
**Start Date**	1 October 2018	14 May 2020	17 June 2019	August 2017	11 January 2013	25 October 2019	1 January 2019
**Number of Births**	80,200	14,100	85,000	315,000	585,000	63,816	163,000 ^α^
**Positives**	2 ^β^	0	0	2 ^γ^	2 ^δ^	0	2
**2TT**	Molecular, tNGS	Molecular, tNGS	None	Biochemical, DBS GAGs ^φ^	Biochemical, DBS GAGs ^∂^	Molecular, tNGS	Molecular, Sanger
**Screen Positives**	58/80,200 births	13/14,100 births	250/85,000 births	780/315,000 births	306/585,000 births	35/63,816 births	652/163,000 births
**Screen Positive rate (%) ^ξ^**	0.072%	0.092%	0.294%	0.248%	0.052%	0.055%	0.400%

^α^ Virginia reported testing 167,969 specimens through 7/31/2020, which included repeat NBS tests. We adjusted this figure to match the birthrate based on the CDC annual birth data for Virginia [21]. Note that the screen positive rate for Virginia includes repeat NBS on newborns that previously screened positive; ^β^ Both cases attenuated; ^γ^ One severe, one unknown; ^δ^ Both severe—Missouri also detected a third case with positive GAGs on 2TT, but parental refusal of confirmatory testing leaves this case unresolved; ^φ^ Michigan started DBS GAG 2TT in December 2017; ^∂^ Missouri started DBS GAG 2TT in March 2020; ^ξ^ Due to the disparity between these programs in their deployment of 2TT, the screen positive rate utilized in this table is the initial rate of presumptive positive NBS before application of 2TT.

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
