# Peer review of "Digital Microfluidics in Newborn Screening for Mucopolysaccharidoses: A Progress Report"

_2409-515X, 2020, doi:10.3390/ijns6040078_

Round 1
Reviewer 1 Report
This is a research report outlining an alternative analytical method for Newborn Screening for Lysosomal Storage Diseases. The NBS for several LSD has been developed in US and is being developed in some countries in Europe however the false positives rate from dry blood spot method is high. The DMF method may provide more robust, more sensitive and specific way of diagnosing new LSDs, in particular MPS I. It may improve the referral pathway for all truly positive cases.
The authors outline its advantages and limitations.
The paper is a well-written and well-structured paper; subheadings make it easy to follow.
Author Response
We thank this reviewer for their positive comments.
We did not see any comment requiring a response.
Reviewer 2 Report
Thank you for giving me the opportunity to review this interesting article by Dr. Washburn and Dr. Millinton. Dr. Millington is a well-known scholar in the newborn screening field for lysosomal storage disorders. The authors provided insightful current status and future plans by using the digital microfluidic method for MPS I diseases. The manuscripts and figures were clearly illustrated and should be able to be accepted after submitting the revision back to the editors and reviewers.
- The video was fascinating and demonstrated the process clearly. The reagents are passing through the cartridge from up to down and the patient samples are passing through right to left. I believe the intersection parts are overlapping. How to avoid cross-contamination when the substrate reagents passing through the area that the patient samples just passing through? Is there any evidence showing that there is no carryover from very high enzyme activity samples to non-enzyme activity samples if they happen to put next to each other?
- Hemoglobin is a well-known interference for the fluorescence assay due to absorbance similarity and it would cause false positives and false negatives in the real practice, but the DMF data showing in this paper looks just fine. Please illustrate how the DMF method addresses this issue compared with the traditional fluorescent method.
- Following the previous question, which may not relevant to this article, but this issue might happen to a lot of newborn screening labs who use DMF as a first-tier method. Washington State has used DMF assay for MPS I and Pompe screening, MPS works fine like the data showing in this paper. However, they got HUGE trouble with very high false-positive rates for Pompe screening. It causes a lot of burden for both newborn screening labs, referring hospitals, and of course, patients and parents. The newborn screening lab needs to send part of the DBS to referring hospitals that use the traditional 4MU method or send the sample to BABIES using genetic tests as a second-tier assay to eliminate the false positives. These processes are laborious and timely insensitive especially for infantile Pompe babies who need treatment right after birth. It will be very helpful if the author can provide the solutions for NBS lab who use DMF as their first-tier LSD screening and have high false-positive rates.
- Does BABIES provide appropriate cutoff values for each disease? Or the newborn screening lab should define the cutoff assay by themselves? In Table 1, the positive rates are 7-8 folds different from each lab (0.052-0.4%). What is the reason causing that? And how these newborn screening lab with high positive rates to address this issue? Will adjusting the cutoff solve this issue? Will the lower cutoff value cause any false-negative results?
- The newborn screening labs in Taiwan are definitely the pioneers in this field and they lead the MPS newborn screening in the world as well. Apparently, there are more than one newborn screening labs in Taiwan who use MSMS for LSDs screening and they began MPS screening much earlier and doing a much better job than ref 27. (https://pubmed.ncbi.nlm.nih.gov/30409495/ and https://doi.org/10.1186/s13023-018-0816-4). If the authors are going to refer and compare the data with MSMS methodology, these data should be more comprehensive.
- GAG was proposed as the second-tier assay to minimize the false positives. However, GAG assay from DBS is ONLY available by LCMSMS assay, which means the labs need to purchase BOTH DMF and LCMS instruments. In this case, what is the benefit of using DMF as the first-tier assay?
- The author may consider include a cost analysis of DMF assay (cost can be different from applying one assay to multiple assays). I believe the cost and the availability of substrates/instrument/maintenance are one of the advantages of using the DMF method and that is one of the key factors that most newborn screening lab directors worry about.
Author Response
Responses to Reviewer 2.
1. The video was fascinating and demonstrated the process clearly. The reagents are passing through the cartridge from up to down and the patient samples are passing through right to left. I believe the intersection parts are overlapping. How to avoid cross-contamination when the substrate reagents passing through the area that the patient samples just passing through? Is there any evidence showing that there is no carryover from very high enzyme activity samples to non-enzyme activity samples if they happen to put next to each other?
Response: This was already addressed on line 105-6 in section 2.1 of the manuscript. We added a comment to the effect that the droplets are not in direct physical contact with the electrode surfaces, thus the carry-over is negligible with an appropriate reference
2. Hemoglobin is a well-known interference for the fluorescence assay due to absorbance similarity and it would cause false positives and false negatives in the real practice, but the DMF data showing in this paper looks just fine. Please illustrate how the DMF method addresses this issue compared with the traditional fluorescent method.
Response: This is addressed in reference 19. We added an explanatory comment in the text in section 2.4.1, L.157-159.
3. Following the previous question, which may not relevant to this article, but this issue might happen to a lot of newborn screening labs who use DMF as a first-tier method. Washington State has used DMF assay for MPS I and Pompe screening, MPS works fine like the data showing in this paper. However, they got HUGE trouble with very high false-positive rates for Pompe screening. It causes a lot of burden for both newborn screening labs, referring hospitals, and of course, patients and parents. The newborn screening lab needs to send part of the DBS to referring hospitals that use the traditional 4MU method or send the sample to BABIES using genetic tests as a second-tier assay to eliminate the false positives. These processes are laborious and timely insensitive especially for infantile Pompe babies who need treatment right after birth. It will be very helpful if the author can provide the solutions for NBS lab who use DMF as their first-tier LSD screening and have high false-positive rates.
Response: This is a manuscript on MPS, so we will not address Pompe disease. The published experience of the Missouri NBS program, who have had the most experience with DMF, may be instructive (see ref. 18) We addressed this in our manuscript by making a recommendation that programs consider using the CLIR post-analytical tool, which now has several adherents, that can significantly mitigate this issue (section 3.2.3, L.266-277).
4. Does BABIES provide appropriate cutoff values for each disease? Or the newborn screening lab should define the cutoff assay by themselves? In Table 1, the positive rates are 7-8 folds different from each lab (0.052-0.4%). What is the reason causing that? And how these newborn screening lab with high positive rates to address this issue? Will adjusting the cutoff solve this issue? Will the lower cutoff value cause any false-negative results?
Response: Babies do not provide cut-off values. They do advise new users to contact the Missouri NBS program, who have had the most experience in the use of DMF, for advice on this.
At this time it is unclear what is causing the discrepancies seen with the presumptive positive NBS rates, and we have addressed this in a general sense in the manuscript by adding further comments in section 3.2.2 (L.241-251) and 3.2.3.
5. The newborn screening labs in Taiwan are definitely the pioneers in this field and they lead the MPS newborn screening in the world as well. Apparently, there are more than one newborn screening labs in Taiwan who use MSMS for LSDs screening and they began MPS screening much earlier and doing a much better job than ref 27. (https://pubmed.ncbi.nlm.nih.gov/30409495/ and https://doi.org/10.1186/s13023-018-0816-4). If the authors are going to refer and compare the data with MSMS methodology, these data should be more comprehensive.
Response: We acknowledge the leading and pioneering role of the Taiwan NBS program. In this article, we did not seek to make a comprehensive comparison with MS/MS – we focused our comparison on programs that use the PE commercial kit for multiple LSDs using MS/MS. Nevertheless, we have added a sentence that includes the references you suggested in section 3.2.2 (L,228-235) and explained why we would like to keep the original reference to the multiplex assay in the text as well.
6. GAG was proposed as the second-tier assay to minimize the false positives. However, GAG assay from DBS is ONLY available by LCMSMS assay, which means the labs need to purchase BOTH DMF and LCMS instruments. In this case, what is the benefit of using DMF as the first-tier assay?
Response: the NBS programs in the US that use this second-tier assay refer their specimens to Mayo Clinic – because they do not have the capability to do this in-house. We added a comment to that effect in section 3.2.3. (L.258-265) We also added a comment that most NBS labs already use MS/MS for acylcarnitine and amino acid NBS, and several are looking at using any extra capacity they might have for second-tier testing, as proposed by Ombrone. et al. (Mass Spectrom Rev. 2016, 35, 71-84). Most of these methods utilize LC-MS/MS, and could add GAG assays if they so desire – since only a limited number of specimens would require such testing on a daily basis. There is no reason to add more MSMS systems specifically for this purpose.
7. The author may consider include a cost analysis of DMF assay (cost can be different from applying one assay to multiple assays). I believe the cost and the availability of substrates/instrument/maintenance are one of the advantages of using the DMF method and that is one of the key factors that most newborn screening lab directors worry about.
We agree with the reviewer, however, cost analysis was not the main objective in this manuscript, although it has been addressed in previous publications. We have added a sentence with a reference to address this in section 2.4.1 (L.153-155).